# Two Efficient Sparse Fourier Algorithms Using the Matrix Pencil Method

**Bin Li, Xueqing Hou, Zhikang Jiang** 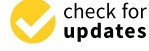 **and Jie Chen** *

School of Mechanical and Electrical Engineering and Automation, Shanghai University, Shanghai 200072, China;
sulibin@shu.edu.cn (B.L.); xueqinghou@163.com (X.H.); zkjiang@i.shu.edu.cn (Z.J.)
* Correspondence: jane.chen@shu.edu.cn

**Abstract:** The research on efficient computation of sparse signals by various Sparse Fast Fourier Transform (sFFT) algorithms has always been a hot topic in the direction of signal processing. The algorithms can decrease the sampling and running complexity by taking advantage of the signal's inherent characteristics that a large number of signals are sparse in the frequency domain. The sFFT algorithm is generally divided into two stages: the first stage is bucketization. The process is to divide $N$ frequencies into $B$ buckets through the filter. The main filters used are the flat window filter and the aliasing filter. The second stage is the spectrum recovery. The process is to successfully locate the position of the large frequency in each bucket and successfully calculate the amplitude. Among these steps, the most difficult and time-consuming step is to successfully locate the position of the large value frequency. For the sFFT algorithm based on a flat window filter, the voting method used in sFFT1.0 and sFFT2.0 algorithms should use many rounds, so these two algorithms are time-consuming and indeterministic, while the phase estimation method used in sFFT3.0 and sFFT4.0 algorithms has medium robustness. For sFFT algorithms based on an aliasing filter, the Prony method used in sFFT-DT1.0 algorithm is only applicable to noiseless signals, while the enumeration method used in the sFFT-DT2.0 algorithm has high complexity and poor robustness. In view of the performance of the old methods, new and more efficient methods are needed to achieve spectrum recovery. The spectrum restoration can be converted to estimating the complex amplitudes and attenuation coefficient in the model of the sum of complex exponentials. The matrix pencil method is a standard technique for mode frequency identification. Therefore, we propose the sFFT5.0 algorithm and sFFT-DT3.0 algorithm using the matrix pencil method to do spectrum recovery. These two algorithms are low computational complexity and strong robustness and have achieved good results in the actual comparative test.

**Keywords:** Sparse Fast Fourier Transform (sFFT); aliasing filter; flat window filter; computational complexity; matrix pencil method

## 1. Introduction

As we all know, the best method of the Discrete Fourier Transform (DFT) operation is the famous Fast Fourier Transform (FFT) [1], but the complexity of FFT is O($N$log$N$), which can not meet the needs of some real-time applications and big data requirments. Therefore, sFFT algorithms are proposed to recover only the large value frequency in the sparse signal by using the sparsity feature of the signal. Once proposed, the new algorithm has attracted wide attention and has been continuously developed. In 2012, it was rated as one of the top 10 breakthrough technologies of the year by MIT Technology Review.

The implementation of the sparse Fourier algorithm mainly consists of two steps: bucketization and spectrum recovery. According to different filters used in bucketization, the algorithms can be divided into three categories [2–5]. The first kind of sFFT algorithm is the randomized algorithm based on the Dirichlet kernel filter bank. The most typical algorithm is the class Ann Arbor Fast Fourier Transform (AAFFT) [6–8] algorithm proposed

by the University of Michigan. Their inefficient method of spectrum location is the binary search technique. The second kind of the sFFT algorithm is based on the flat window filter. The most typical algorithm is the class Sparse Fast Fourier Transform (sFFT) [3,9,10] algorithm proposed by MIT. Their defective methods of spectrum location are the voting method and phase estimation method. The third kind of sFFT algorithm is based on the aliasing window filter. The first typical algorithm is the class Fast Fourier Aliasing-based Sparse Transform (FFAST) [11–13] algorithm proposed by Berkeley University. The third typical algorithm is the class sFFT by downsampling in the time domain (sFFT-DT) [14,15] algorithm proposed by Taiwan University. The third typical algorithm is the class Deterministic sFFT (DSFFT) [16] algorithm proposed by Gottingen University. Their methods of spectrum location are the prony method, phase estimation method, enumeration method, etc. The methods mentioned above for spectrum estimation will be introduced later and compared with the new matrix pencil method we introduced.

The problem of locating the position and estimating the amplitude of the large value frequency in the bucket after bucketization can also be equivalent to the problem of solving the "sum of complex exponentials" signal model. The common method of the current sFFT algorithm is to default to only one effective complex exponential signal in the bucket and then calculate it in their own different ways. Such an assumption is impossible to process when there is more than one effective complex exponential signal for the case of aliasing in the bucket, so multiple rounds of calculation are required to reduce the probability of aliasing. Therefore, we can introduce other methods to solve the problem of the "sum of complex exponentials" signal model to solve the problem of the aliasing bucket.

There are two main methods to deal with the problem of the "sum of complex exponentials" signal model. The first and most commonly used solution is the Prony method and its improved algorithm [17–19]. In 1879, French scientist Prony proposed the famous Prony method. Kumaresan [20] improved the Prony Algorithm and relaxed the order requirements in the signal calculation process. These methods are summarized as polynomial methods. The characteristic of this kind of method is that it needs two steps to solve the poles. The first step is to solve a matrix equation to obtain the coefficients of the polynomial, and then further solve the required poles through the roots of the polynomial. In 1975, when Mittra [21,22] applied the classical Prony method to extract poles from transient signals, they found that this method had poor robustness ability in practice. Therefore, another new spectrum estimation method is proposed. In 1979, Pricc [23] proposed the singular value decomposition (SVD) method, and its robustness ability was improved compared with the Prony method. In 1980, Sarker [24] put forward the matrix pencil method (MPM). Its basic idea is to construct two special data matrices based on the data, and solve their generalized eigenvalues according to the relationship between the data matrices. The generalized eigenvalues contain the required solution information. In this way, the problem of solving the "sum of complex exponentials" signal model is transformed into the problem of solving the generalized eigenvalues of the matrix pencil. In the process of solving generalized eigenvalues, in order to suppress noise interference, singular value decomposition and low-rank approximation of matrix are introduced to make the algorithm have good robustness. Based on this, many algorithms have been produced, such as the TLS-ESPRIT [25] algorithm, Pro-ESPRIT [26] algorithms, and so on. The use of the matrix pencil method to solve the aliasing bucket in sFFT algorithms has also been preliminarily explored. For example, MIT and others have proposed the Matrix Pencil FFT(MPFFT) algorithm [24], but the research on this is only a preliminary stage and needs further research. These further specific works will be covered in this paper.

The paper is structured as follows: Section 2 provides a brief overview of the basic concepts of the sFFT algorithm. Section 3 introduces two bucketization techniques of the sFFT algorithm and the "sum of complex exponentials" signal model solving techniques. In Section 4, the new sFFT5.0 algorithm and new sFFT-DT3.0 algorithm are proposed by using the MPP method. In Section 5, the time performance and robustness performance of

the two new algorithms and other sFFT algorithms are tested. It is proved that the new two algorithms are more efficient and robust.

Through the research of this paper, we can get the advantages and novelty of the new two sFFT algorithms compared with the old algorithms. In theory, the time complexity of the new algorithms is independent of $N$, so they are very suitable for processing large length signals. In addition, they have good robustness due to the use of the new matrix pencil method. These theoretical conclusions have been verified by experiments, and their general performance table has been obtained through this paper, which provides a reference for the use and research of these two new excellent sFFT algorithms.

## 2. Preliminaries

In this section, we introduce the preliminary knowledge and basic definitions of sparse Fourier transform.

The $N$-th root of unify is denoted by $\omega_N = e^{-2\pi \mathbf{i}/N}$. The DFT matrix of size $N$ is denoted by $\mathbf{F}_N \in \mathbb{C}^{N \times N}$ as follows:

$$\mathbf{F}_N[j,k] = \frac{1}{N}\omega_N^{jk} \tag{1}$$

The DFT of a vector $x \in \mathbb{C}^N$ (consider a signal of size $N$) is a vector $\hat{x} \in \mathbb{C}^N$ defined as follows:

$$\hat{x} = \mathbf{F}_N x$$
$$\hat{x}_i = \frac{1}{N}\sum_{j=0}^{N-1} x_j \omega_N^{ij} \tag{2}$$

If the spectrum $\hat{x}$ has exactly $K$ non-zero frequency coefficients while the remaining $N - K$ coefficients are zero, the signal is exactly $K$-sparse. If the spectrum $\hat{x}$ has $K$ significant frequency coefficients while the remaining $N - K$ coefficients are approximately equal to zero, the signal is general $K$-sparse. The goal of sFFT is to recover a $K$-sparse approximation $\hat{x}'$ by locating $K$ frequency positions $f_0, \ldots, f_{K-1}$ and estimating $K$ largest frequency coefficients $\hat{x}_{f_0}, \ldots, \hat{x}_{f_{K-1}}$.

## 3. Techniques

In this section, we introduce five technologies related to the new sFFT algorithm. They are frequency bucketization by the flat window filter, frequency bucketization by the aliasing filter, estimating the number of effective signals of the "sum of complex exponentials" signal model, the traditional Prony method used to calculate the poles of the model, and the new MPP method used to calculate the poles of the model.

### 3.1. Frequency Bucketization by the Flat Window Filter

The first stage of sFFT is encoding by frequency bucketization. The process of frequency bucketization using the flat window filter is achieved through shift operation, scaling operation, flat filter operation, and subsampling operation.

The shift operation representing the original signal multiplied by matrix $\mathbf{S}_\tau$. $\mathbf{S}_\tau \in \mathbb{R}^{N \times N}$ (the offset parameter is denoted by $\tau \in \mathbb{R}$) is defined as Equation (3). The vector $x' = \mathbf{S}_\tau x$, such that $x'_i = x_{(i-\tau)}$. The time scaling operation representing the original signal multiplied by matrix $\mathbf{P}_\sigma \in \mathbb{R}^{N \times N}$ (the scaling parameter is denoted by $\sigma \in \mathbb{R}$) is defined as Equation (4). A vector $x' = \mathbf{P}_\sigma x$, such that $x'_i = x_{\sigma i}$. Let matrix $\mathbf{Q}_L \in \mathbb{C}^{N \times N}$ be a diagonal matrix whose diagonal entries represent flat window filter coefficients (filter $G \in \mathbb{C}^N$ be an $(L/N, L/2N, \delta, w)$ flat window) in the time domain, defined as Equation (5). The last operation is the frequency subsampled operation, and the signal in the time domain is

aliased such that the corresponding signal in the frequency domain is subsampled. Let matrix $\mathbf{U}_L \in \mathbb{R}^{B \times N}$ represent the aliasing operator as Equation (6):

$$\mathbf{S}_\tau[j,k] = \begin{cases} 1, & j - \tau \equiv k (\text{mod} N) \\ 0, & \text{o.w.} \end{cases} \tag{3}$$

$$\mathbf{P}_\sigma[j,k] = \begin{cases} 1, & \sigma j \equiv k (\text{mod} N) \\ 0, & \text{o.w.} \end{cases} \tag{4}$$

$$\mathbf{Q}_L[j,k] = \begin{cases} G_j, & j = k \\ 0, & \text{o.w.} \end{cases} \tag{5}$$

$$\mathbf{U}_L[j,k] = \begin{cases} 1, & j - k \equiv 0 (\text{mod} B) \\ 0, & \text{o.w.} \end{cases} \tag{6}$$

Through the Introduction, we have learned about the above four operations. After the above four operations, the signal can be divided into buckets based on the flat window filter. The specific process is as follows: $\hat{y}_{L,\tau,\sigma} = \mathbf{F}_B \mathbf{U}_L \mathbf{Q}_L \mathbf{S}_\tau \mathbf{P}_\sigma x$. The results of bucketization in bucket *i* are shown in Equation (7). In addition, the specific contents can be seen in paper [3]:

$$\hat{y}_{L,\tau,\sigma}[i] \approx \hat{G}_{\frac{L}{2}} \hat{x}_{\sigma^{-1}\left(\frac{(2i-1)L}{2}\right)} \omega_N^{\tau\left(\frac{(2i-1)L}{2}\right)} + \ldots + \hat{G}_{-\frac{L}{2}+1} \hat{x}_{\sigma^{-1}\left(\frac{(2i+1)L}{2}-1\right)} \omega_N^{\tau\left(\frac{(2i+1)L}{2}-1\right)} \tag{7}$$

### 3.2. Frequency Bucketization by Aliasing Filter

The process of frequency bucketization using the aliasing filter is achieved through shift operation and aliasing filter operation.

The first operation time shift operation is similar to that described in the previous section. The second operation is the use of the aliasing filter. The signal in the time domain is subsampled such that the corresponding spectrum in the frequency domain is aliased. The subsampling factor is denoted by $L$, representing how many frequencies are aliasing in one bucket. The subsampling number is denoted by $B(B = N/L)$. The subsampling operation represents the original signal multiplied by matrix $\mathbf{D}_L$. $\mathbf{D}_L \in \mathbb{R}^{B \times N}$ is defined as follows:

$$\mathbf{D}_L[j,k] = \begin{cases} 1, & k = jL \\ 0, & \text{o.w.} \end{cases} \tag{8}$$

After the above two operations, the signal can be divided into buckets based on the aliasing filter. The specific process is as follows: $\hat{y}_{B,\tau} = \mathbf{F}_B \mathbf{D}_L \mathbf{S}_\tau x$, The results of bucketization in bucket *i* are shown in Equation (9). In addition, the specific contents can be seen in [4]:

$$\hat{y}_{B,\tau}[i] = \sum_{j=0}^{L-1} \hat{x}_{jB+i} \omega^{\tau(jB+i)} \tag{9}$$

### 3.3. Estimation of the Number of Effective Signals

This section describes the first step in solving the "sum of complex exponentials" signal model, estimating the number of effective signals.

#### 3.3.1. Problem Statement

In general, the signal model of the observed late time of electromagnetic-energy-scattered response from an object can be considered as the "sum of complex exponentials" signal model and be described as follows:

$$m_k \approx \sum_{j=0}^{a_m-1} p_j z_j^k \approx \sum_{j=0}^{a-1} p_j z_j^k \tag{10}$$

In the above formula, $m_K$ represents the sampling signal at time $k$, $p_j$ represents a complex coefficient containing amplitude information, $z_j$ represents an attenuation index containing phase information and represents an exponential with an amplitude equal to 1. $a_m$ means that we assume that the sampled signal is composed of the sum of at most $a_m$ effective complex exponential signals. In addition, the sampled signal is actually composed of the sum of $a$ effective complex exponential signals. The approximate equal sign indicates that the signal contains noise in addition to the effective signal. The first problem now is how to estimate the number of effective complex exponential signals.

### 3.3.2. Problem Solution

According to the maximum possible, the sampling signal is composed of at most $a_m$ number of signals, $2a_m$ times is used for sampling. According to the sampled values, a matrix is constructed, SVD decomposition is performed on the matrix, and PCA analysis is performed to obtain the number of effective complex signals.

The specific process is as follows. First, $2a_m$ samples are taken to obtain Equation (11). The matrix $\mathbf{M}_{a_m}$ is constructed as Equation (12) according to $2a_m$ sampling results. In addition, the vector $\hat{z}_j \in \mathbb{C}^{a_m \times 1}$ defined as $\hat{z}_j = [z_j^0, z_j^1, \ldots, z_j^{a_m-1}]^T$ is composed of each attenuation index:

$$
\begin{bmatrix}
z_0^0 & z_1^0 & \cdots & z_{a_m-1}^0 \\
z_0^1 & z_1^1 & \cdots & z_{a_m-1}^1 \\
\cdots & \cdots & \cdots & \cdots \\
z_0^{2a_m-1} & z_1^{2a_m-1} & \cdots & z_{a_m-1}^{2a_m-1}
\end{bmatrix}
\begin{bmatrix}
p_0 \\ p_1 \\ \cdots \\ p_{a_m-1}
\end{bmatrix}
\approx
\begin{bmatrix}
m_0 \\ m_1 \\ \cdots \\ m_{2a_m-1}
\end{bmatrix}
\tag{11}
$$

$$
\mathbf{M}_{a_m} =
\begin{bmatrix}
m_0 & m_1 & \cdots & m_{a_m-1} \\
m_1 & m_2 & \cdots & m_{a_m} \\
\cdots & \cdots & \cdots & \cdots \\
m_{a_m-1} & m_{a_m} & \cdots & m_{2a_m-1}
\end{bmatrix}_{a_m \times a_m}
\tag{12}
$$

From the above three definitions, it can be concluded that the relationship between $\mathbf{M}_{a_m}$, $p_j$, $\hat{z}_j$ satisfies the following Theorem 1:

**Theorem 1.**

$$
\mathbf{M}_{a_m} \approx \sum_{j=0}^{a_m-1} p_j \hat{z}_j \hat{z}_j^T
\tag{13}
$$

**Proof of Theorem 1.**

$$
p_0 \hat{z}_0 z_0^0 + p_1 \hat{z}_1 z_1^0 + \cdots + p_{a_m-1} \hat{z}_{a_m-1} z_{a_m-1}^0 \approx [m_0, m_1, \ldots, m_{a_m-1}]^T
$$

$$
p_0 \hat{z}_0 z_0^1 + p_1 \hat{z}_1 z_1^1 + \cdots + p_{a_m-1} \hat{z}_{a_m-1} z_{a_m-1}^1 \approx [m_1, m_2, \ldots, m_{a_m}]^T
$$

$$
\cdots
$$

$$
p_0 \hat{z}_0 z_0^{a_m-1} + p_1 \hat{z}_1 z_1^{a_m-1} + \cdots + p_{a_m-1} \hat{z}_{a_m-1} z_{a_m-1}^{a_m-1} \approx [m_{a_m-1}, m_{a_m}, \ldots, m_{2a_m}]^T
$$

Based on the properties as mentioned above, we obtain Equation (13). □

Equation (13) is similar to the symmetric singular value decomposition (SSVD). Nevertheless, there are some differences. Ref. [27] proved that, for the symmetric matrix, the $\Sigma$ obtained from the SVD is equal to the $\Sigma$ obtained from the SSVD. For example, the SVD of $\begin{bmatrix} 0 & 1 \\ 1 & 0 \end{bmatrix}$ is $\begin{bmatrix} 0 & 1 \\ 1 & 0 \end{bmatrix} = \begin{bmatrix} 0 & 1 \\ 1 & 0 \end{bmatrix} \begin{bmatrix} 1 & 0 \\ 0 & 1 \end{bmatrix} \begin{bmatrix} 1 & 0 \\ 0 & 1 \end{bmatrix}$ and the SSVD of $\begin{bmatrix} 0 & 1 \\ 1 & 0 \end{bmatrix}$ is $\begin{bmatrix} 0 & 1 \\ 1 & 0 \end{bmatrix} = \frac{1}{\sqrt{2}} \begin{bmatrix} 1 & -i \\ 1 & i \end{bmatrix} \begin{bmatrix} 1 & 0 \\ 0 & 1 \end{bmatrix} \frac{1}{\sqrt{2}} \begin{bmatrix} 1 & 1 \\ -i & i \end{bmatrix}$; the $\Sigma$ values gained via these two methods are the same. After knowing this, we can compute the SVD of $\mathbf{M}_{a_m}$ and obtain $a_m$ singular values, then perform the principal component analysis (PCA), $\sigma_1 \geq \sigma_2 \geq \cdots \geq \sigma_{a_m}$. In

these, $a_m$ number of singular values $\sigma_j$'s, the amount of large singular values indicates the amount of efficient components $p_j$'s.

For a matrix that is not very large, the calculation amount of PCA is the third power of the matrix size, so the time complexity is $a_m^3$, and the sampling complexity is $2a_m$ for the estimation of the number of effective complex exponential signals.

### 3.4. Calculate the Poles by the Prony Method

3.4.1. Problem Statement

After knowing the effective number $a$, the original Equation (12) becomes Equation (14) and the original matrix $\mathbf{M}_{a_m}$ becomes a new matrix $\mathbf{M}_a$. In the next step, it is necessary to estimate each complex coefficient $p_j$ and each attenuation index $z_j$ according to the $m_j$'s:

$$
\begin{bmatrix}
z_0^0 & z_1^0 & \cdots & z_{a-1}^0 \\
z_0^1 & z_1^1 & \cdots & z_{a-1}^1 \\
\cdots & \cdots & \cdots & \cdots \\
z_0^{2a-1} & z_1^{2a-1} & \cdots & z_{a-1}^{2a-1}
\end{bmatrix}
\begin{bmatrix}
p_0 \\
p_1 \\
\cdots \\
p_{a-1}
\end{bmatrix}
\approx
\begin{bmatrix}
m_0 \\
m_1 \\
\cdots \\
m_{2a-1}
\end{bmatrix}
\tag{14}
$$

$$
\mathbf{M}_a =
\begin{bmatrix}
m_0 & m_1 & \cdots & m_{a-1} \\
m_1 & m_2 & \cdots & m_a \\
\cdots & \cdots & \cdots & \cdots \\
m_{a-1} & m_{a_m} & \cdots & m_{2a-2}
\end{bmatrix}_{a \times a}
\tag{15}
$$

3.4.2. Problem Solution by the Prony Method

The Prony method is a method for solving polynomials, so the following polynomial of order $a$ is defined as follows:

$$
P(z) = z^a + c_{a-1}z^{a-1} + \cdots + c_1 z + c_0
\tag{16}
$$

The Prony method has three steps. The first step is to approximate the coefficient $C$ of the polynomial through the formula, the second step is to approximate $z_j$ through the above approximate polynomial, and the third step is to calculate $p_j$.

Let the orthogonal polynomial formula $P(z)$ be defined as Equation (16) and $P(z) \approx 0$. Let Matrix $\mathbf{M}_a \in \mathbb{C}^{a \times a}$ be defined as Equation (15). Let vector $C$ be defined as $C = [c_0, c_1, \ldots, c_{a-1}]^T$. Let vector $M_s$ be defined as $M_s = [-m_a, -m_{a+1}, \ldots, -m_{2a-1}]^T$. The moments' formula satisfies Theorem 2. Through Theorem 2, we can obtain $C \approx (\mathbf{M}_a^{-1})M_s$. Thus, the first step of the Prony method has been completed.

**Theorem 2.** $\mathbf{M_a}C \approx M_s$

$$
\begin{bmatrix}
m_0 & m_1 & \cdots & m_{a-1} \\
m_1 & m_2 & \cdots & m_a \\
\cdots & \cdots & \cdots & \cdots \\
m_{a-1} & m_{a_m} & \cdots & m_{2a-2}
\end{bmatrix}_{a \times a}
\begin{bmatrix}
c_0 \\
c_1 \\
\cdots \\
c_{a-1}
\end{bmatrix}_{a \times 1}
\approx
\begin{bmatrix}
-m_a \\
-m_{a+1} \\
\cdots \\
-m_{2a-1}
\end{bmatrix}_{a \times 1}
\tag{17}
$$

**Proof of Theorem 2.**

$$
c_0 m_0 \approx (p_0 z_0^0 + \ldots p_{a-1} z_{a-1}^0)c_0
$$

$$
\cdots
$$

$$
c_{a-1}m_{a-1} \approx (p_0 z_0^{a-1} + \ldots p_{a-1} z_{a-1}^{a-1})c_{a-1}
$$

$$
\Rightarrow c_0 m_0 + \cdots + c_{a-1}m_{a-1}
$$

$$
\approx p_0(c_0 z_0^0 + \ldots c_{a-1} z_0^{a-1}) + \cdots + p_{a-1}(c_0 z_{a-1}^0 + \ldots c_{a-1} z_{a-1}^{a-1})
$$

$$
\approx (-p_0 z_0^a) + \cdots + (-p_{a-1} z_{a-1}^a) \approx -m_a
$$

The first element of $M_s$ has been proven and other elements of $M_s$ can also be proven. $\square$

The second step of the Prony method is to calculate the coefficients of the polynomial and find the approximate root of the polynomial. When the polynomial is an equation, directly use the method of solving the equation, so the $a$ number of roots of $P(z) = 0$ is the solution of $z_j$'s, such as if $a = 1$, through $P(z) = z + c_0 = 0$, then $z_0 = -c_0$. If $a = 2$, through $P(z) = z^2 + c_1 z + c_0 = 0$, then $z_0 = (-c_1 - (c_1^2 - 4c_0)^{0.5})/2$, $z_1 = (-c_1 + (c_1^2 - 4c_0)^{0.5})/2$. If the polynomial is an approximate equation, $z_j$'s can be solved by solving the eigenvalue of the matrix. The third step of the Prony method is to calculate $p_j$'s. Obviously, $p_j$'s can be obtained by the least square method when Equation (14) is known.

In the first step, the complexity of calculate polynomial coefficients $C \approx (\mathbf{M}_a^{-1})M_s$ is $a_m^3$. In the second step, the complexity of calculate $a$ approximate roots of the polynomial with known coefficients $P(z) = z^a + c_{a-1} z^{a-1} + \cdots + c_1 z + c_0$ is $a_m^3$. In the third step, the computational complexity of the least square method is $a_m^3$. When dealing with noisy data, the Prony method solves the following of polynomials, so the poles obtained are extremely inaccurate. There is a small deviation in resonance frequency and amplitude, about 10 percent to 20 percent, which is very sensitive to noise. A large number of proportional examples are used in the paper [28,29] to prove the low noise immunity of the Prony method.

### 3.5. Calculate the Poles by the MPP Method

The matrix pencil method, like the Prony method, is a standard technique for mode frequency identification for computing the maximum likelihood signal under Gaussian noise and evenly spaced samples. This method only needs one step to directly convert the pole solving problem into solving the generalized characteristics of the prediction matrix. Ref. [24] points out that this method is efficient and has better accuracy. We first assume that the approximate equation sign becomes the equation sign, that is, there is no noise in the sampled signal.

For our problem, let the Toeplitz matrix **Y** be defined as Equation (18), let $\mathbf{Y_1}$ be **Y** with the rightmost column removed and be defined as Equation (19), and let $\mathbf{Y_2}$ be **Y** with the leftmost column removed and be defined as Equation (20). Let the Vandermondee martix $\mathbf{U}_{a+1} \in \mathbb{C}^{(a+1) \times a}$ be defined as Equation (21):

$$\mathbf{Y} = \begin{bmatrix} m_0 & m_{-1} & \cdots & m_{-a} \\ m_1 & m_0 & \cdots & m_{-a+1} \\ \cdots & \cdots & \cdots & \cdots \\ m_a & m_{a-1} & \cdots & m_0 \end{bmatrix}_{(a+1) \times (a+1)} \tag{18}$$

$$\mathbf{Y_1} = \begin{bmatrix} m_0 & m_{-1} & \cdots & m_{-a+1} \\ m_1 & m_0 & \cdots & m_{-a+2} \\ \cdots & \cdots & \cdots & \cdots \\ m_a & m_{a-1} & \cdots & m_1 \end{bmatrix}_{(a+1) \times (a)} \tag{19}$$

$$\mathbf{Y_2} = \begin{bmatrix} m_{-1} & m_{-2} & \cdots & m_{-a} \\ m_0 & m_{-1} & \cdots & m_{-a+1} \\ \cdots & \cdots & \cdots & \cdots \\ m_{a-1} & m_{a-2} & \cdots & m_0 \end{bmatrix}_{(a+1) \times (a)} \tag{20}$$

$$\mathbf{U}_{a+1} = \begin{bmatrix} 1 & 1 & \cdots & 1 \\ z_0 & z_1 & \cdots & z_{a-1} \\ \cdots & \cdots & \cdots & \cdots \\ z_0^a & z_1^a & \cdots & z_{a-1}^a \end{bmatrix}_{(a+1) \times a} \tag{21}$$

It there is no noise in the sampled signal. so the rank of the matrix $\mathbf{Y}$, $\mathbf{Y_1}$, $\mathbf{Y_2}$ is equal to $a$. Then, Equation (22) can be obtained through the Vandermonde decomposition matrix **Y**. For example, $a = 1$, $\mathbf{Y} = \begin{bmatrix} m_0 & m_{-1} \\ m_1 & m_0 \end{bmatrix} = \begin{bmatrix} p_0 & p_0 z_0^{-1} \\ p_0 z_0^1 & p_0 \end{bmatrix} = \begin{bmatrix} 1 \\ z_0 \end{bmatrix} p_0 \begin{bmatrix} 1 & z_0^{-1} \end{bmatrix}$,

and $a = 2, \mathbf{Y} = \begin{bmatrix} m_0 & m_{-1} & m_{-2} \\ m_1 & m_0 & m_{-1} \\ m_2 & m_1 & m_0 \end{bmatrix} = \begin{bmatrix} 1 & 1 \\ z_0 & z_1 \\ z_0^2 & z_1^2 \end{bmatrix} \begin{bmatrix} p_0 & \\ & p_1 \end{bmatrix} \begin{bmatrix} 1 & z_0^{-1} & z_0^{-2} \\ 1 & z_1^{-1} & z_1^{-2} \end{bmatrix}$. Then, Equation (23) can be obtained through the Vandermonde decomposition matrix $\mathbf{Y}_1$. For example, $a = 2, \mathbf{Y}_1 = \begin{bmatrix} m_0 & m_{-1} \\ m_1 & m_0 \\ m_2 & m_1 \end{bmatrix} = \begin{bmatrix} 1 & 1 \\ z_0 & z_1 \\ z_0^2 & z_1^2 \end{bmatrix} \begin{bmatrix} p_0 & \\ & p_1 \end{bmatrix} \begin{bmatrix} 1 & z_0^{-1} \\ 1 & z_1^{-1} \end{bmatrix}$. Then, Equation (24) can be obtained through the Vandermonde decomposition matrix $\mathbf{Y}_2$. For example, $a = 2$,

$$\mathbf{Y}_2 = \begin{bmatrix} m_{-1} & m_{-2} \\ m_0 & m_{-1} \\ m_1 & m_0 \end{bmatrix} = \begin{bmatrix} 1 & 1 \\ z_0 & z_1 \\ z_0^2 & z_1^2 \end{bmatrix} \begin{bmatrix} p_0 & \\ & p_1 \end{bmatrix} \begin{bmatrix} z_0^{-1} & \\ & z_1^{-1} \end{bmatrix} \begin{bmatrix} 1 & z_0^{-1} \\ 1 & z_1^{-1} \end{bmatrix}.$$

$$\mathbf{Y} = \frac{1}{a+1} \mathbf{U}_{a+1} \mathbf{C} \mathbf{U}_{a+1}^* \tag{22}$$

$$\mathbf{Y}_1 = \frac{1}{a+1} \mathbf{U}_{a+1} \mathbf{C} \mathbf{U}_a^* \tag{23}$$

$$\mathbf{Y}_2 = \frac{1}{a+1} \mathbf{U}_{a+1} \mathbf{C} (\mathrm{diag}(z_j)^*) \mathbf{U}_a^* \tag{24}$$

Through Equations (22)–(24), we can obtain $\mathbf{Y}_2 - \lambda \mathbf{Y}_1 = \frac{1}{a+1} \mathbf{U}_{a+1} \mathbf{C} (\mathrm{diag}(z_j)^* - \lambda \mathbf{I}) \mathbf{U}_a^*$, so the set of generalized eigenvalues of $\mathbf{Y}_2 - \lambda \mathbf{Y}_1$ are the $z_j$'s we seek. If the rank $(\mathbf{Y}) = a$, the set of generalized eigenvalues of $\mathbf{Y}_2 - \lambda \mathbf{Y}_1$ is equal to the set of nonzero² ordinary eigenvalues of $(\mathbf{Y}_1^+)\mathbf{Y}_2$. It is most likely that, for the rank $(\mathbf{Y}) < a$, it is necessary to compute the SVD of the $\mathbf{Y}$, $\mathbf{Y} = \widetilde{\mathbf{V}} \Sigma \mathbf{V}^*$, and then we can use the matrix pencil method to deal with the matrix $\mathbf{V}$ afterward. For details, please refer to paper [24,30]. The third step to calculate $p_j$'s of the MPP method is the least square method similar to the way the Prony method does.

In the first step, the complexity of computing the SVD of the $\mathbf{Y}$, $\mathbf{Y} = \widetilde{\mathbf{V}} \Sigma \mathbf{V}^*$ is $a_m^3$. In the second step, the complexity of calculating the set of nonzero² ordinary eigenvalues of $(\mathbf{Y}_1^+)\mathbf{Y}_2$ is $a_m^3$. In the third step, the computational complexity of the least square method is $a_m^3$.

## 4. Two New sFFT Algorithms

### 4.1. sFFT5.0 Algorithm

The new sFFT5.0 algorithm is a sparse Fourier algorithm based on a flat window filter using the matrix pencil method. The block diagram of this algorithm is shown in Figure 1.

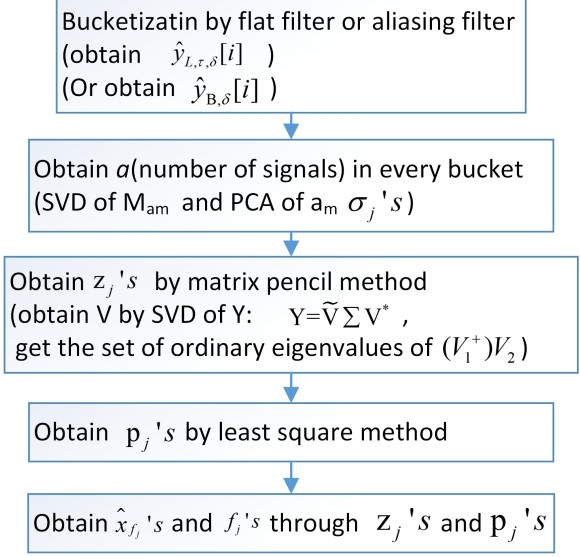

**Figure 1.** The block diagram of two new algorithms.

### 4.1.1. Problem Statement

As shown in Equation (7), the filtered signal of each bucket after the flat window filter is obtained by summing the linear changes of $L$ original signals. In order to recover the spectrum, we need to successfully recover the effective $a$ large value frequency, so Equation (7) is converted to the following equation:

$$
\hat{y}_{L,\tau,\sigma}[i] \approx \sum_{j=0}^{a-1} \hat{G}_{iL-u_j} \hat{x}_{\sigma^{-1}u_j} \omega_N^{\tau u_j} \text{ for } u_j \in [\frac{(2i-1)L}{2}, \frac{(2i+1)L}{2} - 1]
$$

$$
m_k \approx \sum_{j=0}^{a-1} p_j z_j^k
$$

(25)

### 4.1.2. Old Problem Solutions

The old method assumes that there is only one valid signal at most for each bucket of sampling, so the formula can be converted as follows:

$$
\hat{y}_{L,\tau,\sigma}[i] \approx \hat{G}_{iL-u} \hat{x}_{\sigma^{-1}u} \omega_N^{\tau u} \text{ for } u \in [\frac{(2i-1)L}{2}, \frac{(2i+1)L}{2} - 1] \tag{26}
$$

The sFFT1.0 [9] and sFFT2.0 [9] algorithms use the voting method to recover the spectrum of the large value frequency in the bucket. The method is as follows: if $\hat{y}_{L,\tau,\sigma}[i]$'s are large, it means that positions of the effective spectrum may appear in the frequency elements that are hashed by these buckets, then the corresponding position is voted and scored. Correspondingly, if $\hat{y}_{L,\tau,\sigma}[i]$'s are small, it means that positions of the effective spectrum do not exist in the frequency elements that are hashed by these buckets, then the corresponding position is voted and without scoring. After multiple rounds of voting, the position with more vote scores is judged as the position of the effective frequency point. This method is relatively simple. However, due to the more rounds of voting, the operation efficiency will be affected. Moreover, this algorithm is a statistical algorithm, a probabilistic algorithm, not a deterministic algorithm.

The sFFT3.0 [10] and sFFT4.0 [10] algorithms use the phase offset method to recover the spectrum of the large value frequency in the bucket. The method is as follows: for the exactly sparse case, in the first round, we set $\tau_1 = 0$, and, in the second round, we set $\tau_2 = 1$; then, suppose the bucket $i$ contains only one large frequency, so we obtain $\hat{y}_{L,0,\sigma}[i] \approx \hat{G}_{iL-u} \hat{X}_{\sigma^{-1}(u)} \omega_N^{0 \cdot (u)}$ and $\hat{y}_{L,1,\sigma}[i] \approx \hat{G}_{iL-u} \hat{X}_{\sigma^{-1}(u)} \omega_N^{1 \cdot (u)}$, then $\omega_N^{1 \cdot (u)} \approx \frac{\hat{y}_{L,0,\sigma}[i]}{\hat{y}_{L,1,\sigma}[i]}$ so we can locate the position $f = (\sigma^{-1}u) \bmod N$. In a general sparse case, we can estimate the possible range of $u$, narrow the range through multiple iterations, such as binary search or multi-scale search, and finally determine $u$ and $f = (\sigma^{-1}u) \bmod N$. This method has three limitations. One limitation is that the efficiency is very low. The scope needs to be continuously reduced each time in one round, and a total of many rounds are required. The second limitation is that the search has an assumption that only at most one large value in the bucket can be estimated correctly. The third limitation is the general robustness of the algorithm.

### 4.1.3. New Problem Solution

The sFFT5.0 algorithm, as shown in Equation (23), can convert the spectrum problem in one bucket into the solution of the "sum of complex exponentials" signal model. In the solution process, for specified bucket $i$, the number of large value frequencies is defined as $a$, the maximum number of large value frequencies is defined as $a_m$, $\hat{G}_{iL-u_j} \hat{x}_{\sigma^{-1}u_j} = p_j$ represents a complex coefficient, $\omega_N^{u_j} = z_j$ represents attenuation index, $\tau = k$ represents time coefficient, $\hat{y}_{L,\tau,\sigma}[i] = m_k$ represents sampling signal, and $u_j \in [\frac{(2i-1)L}{2}, \frac{(2i+1)L}{2} - 1]$ represents the possible range. After such a transformation, the original problem of the

spectrum restoration in a bucket can be transformed into the problem of solving the "sum of complex exponentials" signal model.

In the actual solution, we assume that there are at most two coefficients in one bucket that means $a_m = 2$. The detail process is as follows: (1) It is known that the maximum number of effective signals is $a_m = 2$. Calculate the number of effective signals $a$ by SVD decomposition and PCA analysis. (2) After the number of signals $a$ is obtained, $z_j$'s are obtained by using the matrix pencil method. (3) $p_j$'s are obtained by using the least square method. After $a$ effective signals are obtained, it can be concluded that if other signals except these $a$ effective signals still have a lot of energy, that is, the maximum number of effective signals representing this bucket is greater than two. Therefore, the bucket division is inconsistent with our preset assumption, so the calculation results are not adopted. These effective signals will be obtained in the case of bucket division in subsequent iterations. (4) Through $\hat{G}_{iL-u_j}\hat{x}_{\sigma^{-1}u_j} = p_j$, $\omega_N^{\tau u_j} = z_j$, $f = (\sigma^{-1}u)\mathrm{mod}N$, obtain every $f_j$ and $\hat{x}_{f_j}$ in one bucket. After the spectrum recovery of all buckets is processed, the spectrum recovery of the signal is completed as well.

### 4.1.4. Performance Analysis in Theory

First, we analyze the time complexity: in one round, the time complexity of the old sFFT1.0 and sFFT2.0 algorithms for each bucket is $O(1)$, a total of $B$ buckets are processed, so the time complexity for spectrum recovery is $O(B)$, the time complexity of previous bucketization processing is $O(B \log B)$, a total of $\log N$ rounds, so the total complexity is at least $O((B \log B + B) \log N) = O(K \log K \log N)$. For the old sFFT4.0 algorithm, the rounds are $O(\log L) = O(\log N/K)$, so the total complexity is at least $O(K \log K \log N/K)$. In one round, the time complexity of the new sFFT5.0 algorithm for each bucket is $O(a_m)$; a total of $B$ buckets are processed, so the time complexity for spectrum recovery is $O(B)$, the time complexity of previous bucketization processing is $O(B \log B)$, a total of $O(a_m) * O(\log K)$ rounds, so the total complexity is at least $O((B \log B + B) \log K) = O(K \log K \log K)$; this complexity has no association with $N$.

Secondly, we analyze the robustness: from the above analysis, the old method assumes that the sampling of each bucket has at most only one effective signal, so it can not deal with aliasing. The new method can handle the case that there are two effective signals in the bucket, and the matrix pencil method ensures that the position and amplitude of one or two large frequency values can be normally estimated even if the bucket length $L$ is large, so the robustness of the new method is better.

### *4.2. sFFT-DT3.0 Algorithm*

The new sFFT-DT3.0 algorithm is a sparse Fourier algorithm based on the aliasing filter using the matrix pencil method. The block diagram of this algorithm is shown in Figure 1.

### 4.2.1. Problem Statement

As shown in Equation (9), the filtered signal of each bucket after the aliasing window filter is obtained by summing the linear changes of $L$ original signals. In order to recover the spectrum, we need to successfully recover the effective $a$ large value frequency, so Equation (9) is converted to the following equation:

$$\hat{y}_{B,\tau}[i] \approx \sum_{j=0}^{a-1} \hat{x}_{f_j} \omega_N^{\tau(f_j)} f_j \in \{i, B+i, 2B+i, \ldots, (L-1)B+i\}$$

$$m_k \approx \sum_{j=0}^{a-1} p_j z_j^k$$

(27)

### 4.2.2. Old Problem Solutions

In the solution process, for a specified bucket $i$, the number of large value frequencies is defined as $a$, the maximum number of large value frequencies is defined as $a_m$, $\hat{x}_{f_j} = p_j$

represents a complex coefficient, $\omega_N^{f_j} = z_j$ represents an attenuation index, $\tau = k$ represents time coefficient, $\hat{y}_{B,\tau}[i] = m_k$ represents a sampling signal, $f_j \in \{i, B+i, 2B+i, \ldots, and(L-1)B+i\}$ represents the possible range. After such a transformation, the original problem of the spectrum restoration in a bucket can be transformed into the problem of solving the "sum of complex exponentials" signal model.

After the number of effective signals $a$ is obtained, the polynomial Prony method is used to continue the solution in the old algorithm. The sFFT-DT1.0 [14,15] algorithm defaults that the polynomial is an equation and uses the polynomial root method to solve it. This is only suitable for the case without noise. The sFFT-DT2.0 [14,15] algorithm tries every possible position $z_j = \omega_N^{f_j}, f_j \in \{i, B+i, 2B+i, \ldots, (L-1)B+i\}$ in Equation (16) $P(z) = z^a + c_{a-1}z^{a-1} + \cdots + c_1 z + c_0$, the first $a$ number of the smallest $|P(z)|$ is the solution of $z_j$'s. $L$ attempts are needed in one solution.

### 4.2.3. New Problem Solution

In the actual solution, we assume that there are at most four coefficients in one bucket that means $a_m = 4$. The detailed process is as follows: (1) It is known that the maximum number of effective signals is $a_m = 4$. Calculate the number of effective signals $a$ by SVD decomposition and PCA analysis. (2) After the number of signals $a$ is obtained, $z_j$'s are obtained by using the matrix pencil method. (3) $p_j$'s are obtained by using the least square method. After $a$ effective signals obtained, it can be concluded that, if other signals except these $a$ effective signals still have a lot of energy, that is, the maximum number of effective signals representing this bucket is greater than four. Therefore, the bucket division is inconsistent with our preset assumption, so the calculation results are not adopted. These effective signals will be obtained in the case of bucket division in subsequent iterations. (4) Through $\hat{x}_{f_j} = p_j, \omega_N^{f_j} = z_j$, get every $f_j$ and $\hat{x}_{f_j}$ into one bucket. After the spectrum recovery of all buckets is processed, the spectrum recovery of the signal is completed as well.

### 4.2.4. Performance Analysis in Theory

The sFFT-DT1.0 algorithm is only suitable for ideal signals by using the polynomial root method, and is not suitable for general situations. The sFFT-DT2.0 algorithm uses the enumeration method. The time complexity of calculating a bucket is $O(L + a_m)$, which is relatively high and has poor robustness. The new sFFT-DT3.0 algorithm uses the matrix pencil method. The time complexity of calculating a bucket is $O(a_m)$, which realizes the optimization of performance and is more robust than the traditional Prony method.

## 5. Algorithms Analysis in Practice

In this section, we evaluate the performance of five sFFT algorithms using the flat filter: the sFFT1.0, sFFT2.0, sFFT3.0, sFFT4.0, and sFFT5.0 algorithms. In addition, we evaluate the performance of five sFFT algorithms using the aliasing filter: the sFFT-DT1.0, sFFT-DT2.0, sFFT-DT3.0, R-FFAST, and DSFFT algorithms as well. These old algorithms are obtained through open source, and the new algorithms are implemented according to the design. There are three main test items, namely, the complexity of the algorithm under different lengths, the complexity of the algorithm under different sparsity, and the $L_1$ error of the algorithms. All experiments were run on a Linux CentOS computer with a 4 Intel(R) Core(TM) i7-4570 3 GHz CPU and 16 GB of RAM.

### 5.1. Experimental Setup

In the experiment, the test signals were gained in a way that $K$ frequencies were randomly selected from $N$ frequencies and assigned a magnitude of 1 and a uniformly random phase. The remaining frequencies were set to zero in the exactly sparse case or combined with additive white Gaussian noise in the general sparse case, whose variance varies depending on the SNR required. The parameters of these algorithms were chosen so

that they can make a balance between time efficiency and robustness. In each experiment, the platform could generate a signal with SNR, $K$, or $N$ as required. The prepared signal and the value of $N$ and $K$ were transmitted to different algorithm libraries through a standard interface. Each test record contains the run time and the $L_0, L_1, L_2$ error between the calculation result and the best result through the algorithm library. Detailed reports can be found on the github website (https://github.com/zkjiang/-/tree/master/docs, accessed on 10 June 2022).

### 5.2. Comparison Experiments of Two New Algorithms Using Different Parameters

In order to better compare and verify the algorithms, this section tests the different parameters of the two proposed algorithms, and determines the selected parameters through the experimental results. Since both algorithms use the matrix pencil method, the most important parameter of the algorithms is the assumed number of maximum effective signals $a_m$ after bucketization. It is obvious that, when $a_m$ is large, the rank of the matrix to be processed is large, so the time complexity will be greatly increased, but, at the same time, the robustness will be improved because it considers the worst possible condition. From Figures 2 and 3, based on the balance of the two performances, the parameter used in our proposed sFFT5.0 algorithm is $a_m = 2$, and the parameter used in sFFT-DT3.0 algorithm is $a_m = 4$.

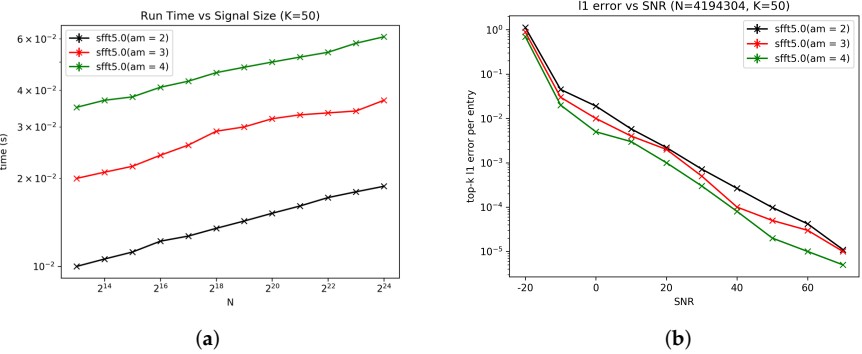

(**a**)  (**b**)

**Figure 2.** Run time and $L_1$-error of sFFT5.0 algorithms using different parameters. (**a**) run time vs. signal size; (**b**) $L_1$-error vs. SNR.

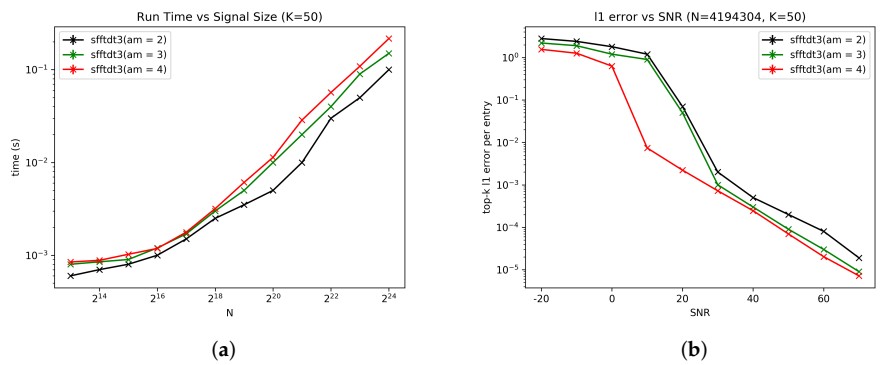

(**a**)  (**b**)

**Figure 3.** Run time and $L_1$-error of of sFFT-DT3.0 algorithms using different parameters. (**a**) Run time vs. signal size; (**b**) $L_1$-error vs. SNR.

### 5.3. Comparison Experiment of sFFT5.0 and Other Algorithms Using the Flat Filter

We plot Figure 4a,b representing the run time vs. signal size and vs. signal sparsity for the sFFT1.0, sFFT2.0, sFFT4.0, and sFFT5.0 algorithms in the general sparse case (The sFFT3.0 algorithm is ignored because it is not suitable for a general sparse case; it is only suitable for an exactly sparse case). Thus, from Figure 4, we can see the following: (1) The run time of these four algorithms were approximately linear in the log scale as a

function of $N$ and nonlinear as a function of $K$. (2) When $N$ is large, the result of ranking run time of the four algorithms is sFFT5.0 > sFFT2.0 > sFFT4.0 > sFFT1.0. From this ranking, it can be seen that, when $N$ is large, the sFFT5.0 algorithm using the matrix pencil method is the best because the complexity of the method is independent of $N$, and other algorithms are related to $N$. (3) When $K$ is large, the result of ranking run time of the four algorithms is sFFT4.0 > sFFT2.0 > sFFT1.0 > sFFT5.0. From this ranking, we can see that, when $k$ is large, the sFFT5.0 algorithm using the matrix pencil method has no advantage. Once there are many buckets to be processed, the matrix transformation involved in each bucket is time-consuming. In addition, through the intersection of these curves, we can also find that, with the increase of sparsity, the time complexity of sFFT5.0 algorithm is better than other algorithms when $N$ is greater than or equal to $2^{22}$ and, with the increase of sparsity, the time complexity of sFFT4.0 algorithm is always better than other algorithms when $K$ is greater than or equal to 100.

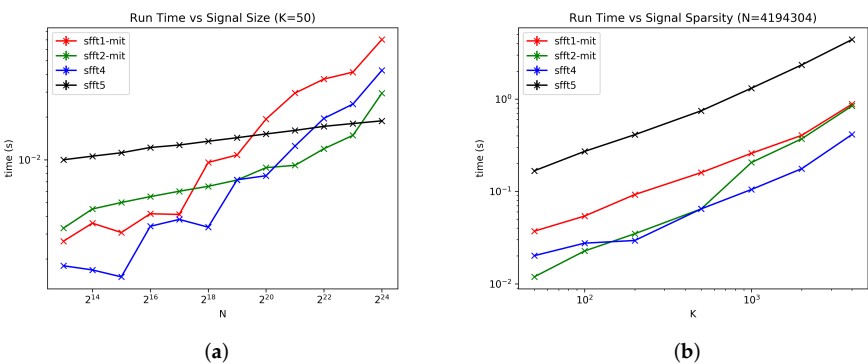

(a)  (b)

**Figure 4.** The run time vs. signal size and vs. signal sparsity for the sFFT1.0, sFFT2.0, sFFT4.0, and sFFT5.0 algorithms in the general sparse case. (**a**) run time vs. signal size; (**b**) run time vs. signal sparsity.

We plot Figure 5a,b representing the run time and $L_1$-error vs. SNR for the sFFT1.0, sFFT2.0, sFFT4.0, and sFFT5.0 algorithms in the general sparse case. Thus, from Figure 5, we can see the following: (1) The runtime is approximately equal vs SNR. (2) To a certain extent, these four algorithms are all robust. (3) The result of ranking robust of the four algorithms is sFFT5.0 > sFFT2.0 > sFFT1.0 > sFFT4.0. The reason is that, in the positioning methods, the matrix pencil method is better than the voting method, and the voting method is better than the phase method, which leads to the advantages and disadvantages of the algorithms using these methods.

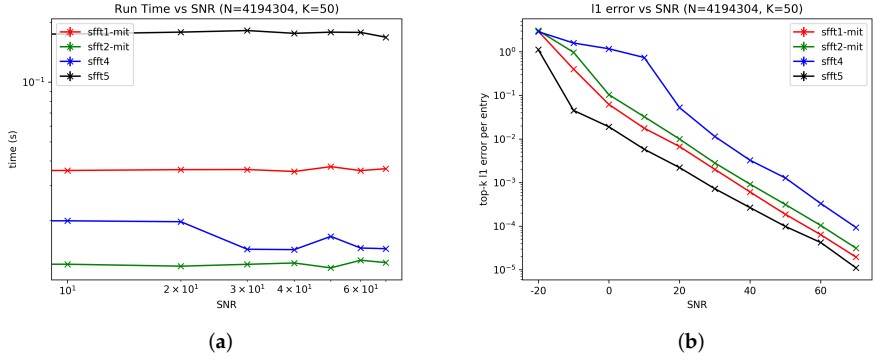

(a)  (b)

**Figure 5.** The run time and $L_1$-error vs. SNR. for the sFFT1.0, sFFT2.0, sFFT4.0 and sFFT5.0 algorithms in the general sparse case. (**a**) run time vs. SNR; (**b**) $L_1$-error vs. SNR.

### 5.4. Comparison Experiment of sFFT-DT3.0 and Other Algorithms Using the Aliasing Filter

We plot Figure 6a,b (The DSFFT algorithm under the condition of large sparsity is ignored because it cannot deal with large sparse data) representing the run time vs. signal size and vs. signal sparsity for the sFFT-DT2.0, sFFT-DT3.0, DSFFT, and R-FFAST algorithm in the general sparse case (The FFAST algorithm and sFFT-DT1.0 algorithm are ignored because they are not suitable for a general sparse case; it is only suitable for an exactly sparse case). Thus, from Figure 6, we can see the following: (1) The run time of these four algorithms were approximately linear in the log scale as a function of $N$ and nonlinear as a function of $K$. (2) The result of ranking run time of the four algorithms is sFFT-DT3.0 > sFFT-DT2.0 > DSFFT > R-FFAST. (3) From this ranking, it can be seen that, in terms of time complexity, the sFFT-DT algorithm using the "sum of complex exponentials" model is the best. Other FFAST algorithms using the peeling framework and DSFFT algorithm using a binary tree framework are more complex and time-consuming. (4) Among the sFFT-DT algorithms, the sFFT-DT3.0 algorithm using the matrix pencil method is better than the sFFT-DT2.0 algorithm using the Prony method because it requires less matrix processing times, and it is independent of $N$, while other algorithms are related to $N$.

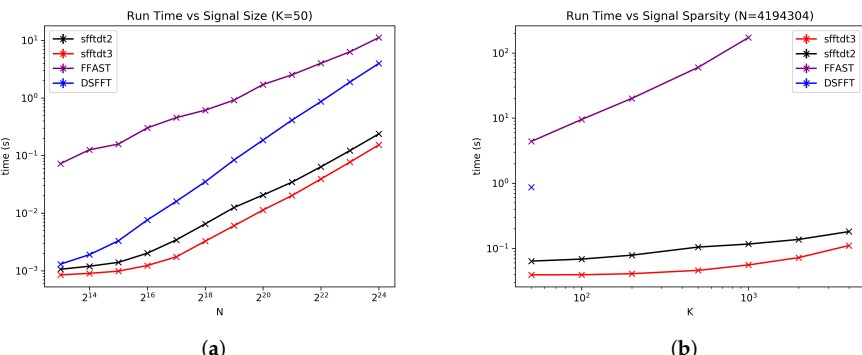

|     |     |
| :-: | :-: |
| (**a**) | (**b**) |

**Figure 6.** The run time vs. signal size and vs. signal sparsity for the sFFT-DT2.0, sFFT-DT3.0, DSFFT, and R-FFAST algorithm in the general sparse case. (**a**) run time vs. signal size; (**b**) run time vs. signal sparsity.

We plot Figure 7a,b representing the run time and $L_1$-error vs. SNR. for the sFFT-DT2.0, sFFT-DT3.0, and R-FFAST algorithm in the general sparse case. Thus, from Figure 7, we can see the following: (1) The runtime is approximately equal vs. SNR except the R-FFAST algorithm. (2) To a certain extent, these three algorithms are all robust. (3) The result of ranking robust of the four algorithms is R-FFAST > sFFT-DT3.0 > sFFT-DT2.0. From this ranking, it can be seen that the R-FFAST algorithm using the peeling framework and MMSE estimation method has the best robustness, but among the sFFT-DT algorithms using the "sum of complex exponentials" model, the sFFT-DT3.0 algorithm is better than the sFFT-DT2.0 algorithm using the Prony method, which is determined by the characteristics of the method used.

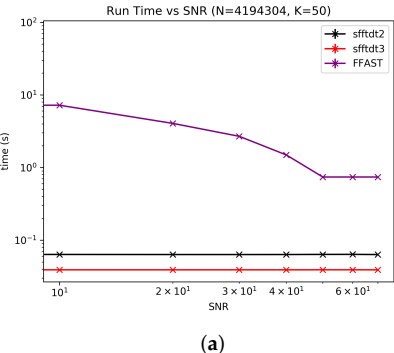
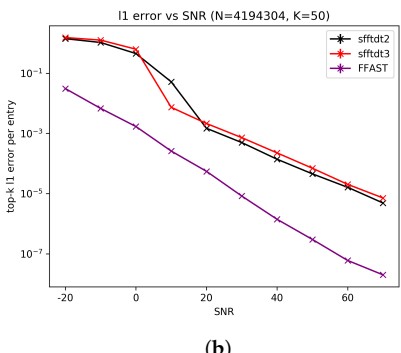

|            |            |
| :--------: | :--------: |
| (**a**)    | (**b**)    |

**Figure 7.** The run time and $L_1$-error vs. SNR for the sFFT-DT2.0, sFFT-DT3.0, and R-FFAST algorithm in the general sparse case. (**a**) run time vs. signal size; (**b**) $L_1$-error vs. SNR.

*5.5. Summary of the Performance of the Proposed Algorithms and the Related Algorithms*

Through theoretical analysis and experimental results, we get a summary table as shown in Table 1.

**Table 1.** The performance of two new algorithms and other sFFT algorithms

| Algorithm | Runtime Complexity in Theory | Rank in Large Length | Rank in Large Sparsity | Rank of Robustness |
| --- | --- | --- | --- | --- |
| sFFT3.0 | $O(K \log N)$ | untested | untested | untested |
| sFFT-DT1.0 | $O(K \log K)$ | untested | untested | untested |
| FFAST | $O(K \log K)$ | untested | untested | untested |
| sFFT1.0 | $O(K^{\frac{1}{2}} N^{\frac{1}{2}} \log^{\frac{3}{2}} N)$ | 4 | 3 | 3 |
| sFFT2.0 | $O(K^{\frac{2}{3}} N^{\frac{1}{3}} \log^{\frac{4}{3}} N)$ | 2 | 2 | 2 |
| sFFT4.0 | $O(K \log N \log_8(N/K))$ | 3 | 1 | 4 |
| sFFT5.0 | $O(K \log K \log K)$ | 1 | 4 | 1 |
| R-FFAST | $O(K \log^{7/3} N)$ | 4 | 3 | 1 |
| DSFFT | $O(K \log K \log N)$ | 3 | 4 | 4 |
| sFFT-DT2.0 | $O(K \log K + N)$ | 2 | 2 | 3 |
| sFFT-DT3.0 | $O(K \log K)$ | 1 | 1 | 2 |

## 6. Conclusions

At the beginning, the paper that introduces the filtered signal is approximately equal to the linear addition of signals in the bucket after aliasing filters or after flat window filters. Based on this situation, the problem of frequency recovery of large values in the bucket is transformed into the problem of solving the amplitude and phase of the "sum of complex exponentials" model. This paper then introduces two methods to calculate the amplitude and phase of the model, the Prony method and MPP method. After comparison, it can be concluded that the MPP method has higher efficiency and stronger robustness.

Based on these, this paper proposes the sFFT5.0 algorithm based on the flat filter and sFFT-DT3.0 algorithm based on the aliasing filter using the matrix pencil method. Compared with the previous sFFT1.0 and sFFT2.0 algorithms using the voting method, the sFFT5.0 algorithm reduces rounds and improves the algorithm efficiency. Compared with the previous sFFT3.0 and sFFT4.0 algorithms using the phase search method, the sFFT5.0 algorithm also reduces rounds, improves the algorithm efficiency, and improves the robustness of the algorithm. Compared with the former sFFT-DT1.0 algorithm using the Prony method, the sFFT-DT3.0 algorithm can process general sparse signals. Compared with the former sFFT-DT2.0 using the enumeration method, the sFFT-DT3.0 algorithm reduces the number of calculations in the bucket, improves the efficiency of the algorithm, and improves the robustness of the algorithm.

Finally, the time complexity and robustness of the new and old four sparse Fourier algorithms based on flat window filter, and the new and old four sparse Fourier algorithms based on aliasing filter are tested. It can be seen from the test results that the two new

algorithms using the matrix pencil method have the best time complexity in the case of large length because the complexity of the method using the matrix pencil method is independent of $N$, and other algorithms are related to $N$. However, in the case of large sparsity, the time complexity of new algorithms is ordinary because, once there are many buckets to be processed, the matrix transformation involved in each bucket of this method is time-consuming. In addition, in terms of robustness, the two new algorithms are also optimal or suboptimal, which is determined by the anti-interference characteristics of the matrix pencil method. These conclusions also prove the superiority of the new algorithm and lay a foundation for the future use of this algorithm.

**Author Contributions:** Conceptualization, Z.J.; methodology, Z.J.; software, Z.J.; validation, Z.J.; formal analysis, Z.J.; investigation, Z.J.; resources, B.L.; data curation, Z.J.; writing—original draft preparation, Z.J.; writing—review and editing, B.L.; visualization, Z.J.; supervision, Z.J.; project administration, X.H.; funding acquisition, J.C. All authors have read and agreed to the published version of the manuscript.

**Funding:** This work was supported by the Youth Program of the National Natural Science Foundation of China under Grant No. 61703263.

**Data Availability Statement:** The datasets used and/or analyzed during the current study are available from the corresponding author on reasonable request. We also made it public on the website https://github.com/zkjiang/-/tree/master/docs. (accessed on 10 June 2022).

**Conflicts of Interest:** The authors declare no conflict of interest.

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
