# Peer review of "Two Efficient Sparse Fourier Algorithms Using the Matrix Pencil Method"

_electronics, doi:10.3390/electronics11152291_

Round 1

Reviewer 1 Report

The authors have presented two sparse Fourier algorithms using matrix pencil method. Technical aspects and result section of the paper should be improved. The following modifications should be considered in the manuscript.

§  Kindly add a paragraph at the end of the introduction section to provide a comparison of the proposed algorithms and emphasize the advantages and the novelties of the proposed algorithms.

§  It is better to write the website address in line 355 as a citation.

§  The figure 1a is plotted as function of “N”, but indicated with “n” in the figure, which should be corrected.

§  From figure 1a, it can be concluded that the runtime of the “sFFT5.0” which is the proposed algorithm by authors are worse compare to the other algorithms in most of cases. Also, from figure 1b, in all cases of k, it can be seen that the proposed sFFT5 algorithm has the worst runtime.

§  The experimental results should be provided with different parameters of the proposed algorithms, for better comparison and clarification.

§  The quality of figures should be improved.

§  It is suggested to provide a block diagram system for the proposed algorithms to improve the readability of the manuscript.

§  Kindly add a table to compare the performance of the proposed algorithms and the related algorithms.

§  Add more results of the proposed algorithms to improve the technical aspects of the manuscript, for example Run time vs. SNR Plot can be added for the proposed algorithms.

Author Response

Response to Reviewer 1 Comments

We would like to thank the reviewers for giving us constructive suggestions which would help us both in English and in depth to improve the quality of the paper. Here we submit a new version of our manuscript with the title “Two Efficient Sparse Fourier Algorithms Using Matrix Pencil Method”, which has been modified according to the reviewers’ suggestions. Efforts were also made to correct the mistakes and improve the English of the manuscript.

General Comments

Point 1: Kindly add a paragraph at the end of the introduction section to provide a comparison of the proposed algorithms and emphasize the advantages and the novelties of the proposed algorithms.

Response 1: Thank the reviewer for the comments. We have added a paragraph to the introduction section, as follows. Through the research of this paper, we can get the advantages and novelty of the new two sFFT algorithms compared with the old algorithms. In theory, the time complexity of the new algorithms is independent of $N$, so they are very suitable for processing large length signals. In addition, they have good robustness due to the use of the new matrix pencil method. These theoretical conclusions have been verified by experiments, and their general performance table has been obtained through this paper, which provides a reference for the use and research of these two new excellent sFFT algorithms.

Point 2: It is better to write the website address in line 355 as a citation.

Response 2: Thank the reviewer for the comments. Your suggestion is greatly appreciated. We agree and therefore the correction has been made in corresponding line.

Point 3: The figure 1a is plotted as function of “N”, but indicated with “n” in the figure, which should be corrected.

Response 3: Thank the reviewer for the comments. In the figure, we changed the symbol of the title and X-axis coordinates from “n” to “N” and from “k” to “K”.

Point 4: From figure 1a, it can be concluded that the runtime of the “sFFT5.0” which is the proposed algorithm by authors are worse compare to the other algorithms in most of cases. Also, from figure 1b, in all cases of k, it can be seen that the proposed sFFT5.0 algorithm has the worst runtime.

Response 4: Thank the reviewer for the comments. Your point of view does reflect the experimental results. For sFFT algorithms, they mainly deals with signals with large data length. Therefore, although sFFT5.0 algorithm has no advantage over other algorithms in dealing with large sparsity signals or small length signals. It also has advantages in dealing with small sparsity and large length signals, so it can still be applied to many big data cases with strong robustness.

Point 5: The experimental results should be provided with different parameters of the proposed algorithms, for better comparison and clarification.

Response 5: Thank the reviewer for the comments. The most important parameter of the two ne new algorithms is the maximum number of large value frequencies am. In Chapter 5, this paper tests the time complexity and robustness of the two new algorithms using different parameters. When am is large, the rank of the matrix to be processed is large, so the time complexity will be greatly increased, but at the same time, the robustness will be improved. Based on the balance of the two performances, the parameter used in our proposed sFFT5.0 algorithm is am = 2, and the parameter used in sFFT-DT3.0 algorithm is am = 4.

Point 6: The quality of figures should be improved.

Response 6: Thank the reviewer for the comments. Your suggestion is greatly appreciated. All figures have been rebuilt for the better display quality.

Point 7: It is suggested to provide a block diagram system for the proposed algorithms to improve the readability of the manuscript.

Response 7: Thank the reviewer for the comments. Your suggestion is greatly appreciated. In Chapter 4, this paper adds the block diagram of two new algorithms.

Point 8: Kindly add a table to compare the performance of the proposed algorithms and the related algorithms.

Response 8: Thank the reviewer for the comments. Your suggestion is greatly appreciated. In Chapter 5, a summary table to compare the performance of the proposed algorithms and the related algorithms has been added.

Point 9: Add more results of the proposed algorithms to improve the technical aspects of the manuscript, for example Run time vs. SNR Plot can be added for the proposed algorithms.

Response 9: Thank the reviewer for the comments. In Chapter 5, we add the Run time vs. SNR experimental results of the proposed algorithms.

Reviewer 2 Report

The article presents studies on the efficient calculation of sparse signals using various sparse Fast Fourier Transform algorithms. Methods and their comparative analysis are presented. There are small questions for the authors.

1. For the described algorithms, we ask the authors to give their decoding of the abbreviation of the name (for example, AAFFT, FFAST, DSFFT and others)

2. The authors use various algorithms such as sFFT1.0, sFFT2.0, sFFT3.0, sFFT4.0, sFFT5.0, and there is confusion in the presented material in the article. Is it possible for the authors to indicate in lines 231 and 242 references to the relevant literature.

3. Is it possible for the authors to create a table of comparative representation of the formulas of all presented algorithms?

4. Why in chapter 5.1. Experimental Setup missing comparison with sFFT3.0 algorithm ?

5. Drawings 1 and 2 should be rebuilt due to the fact that they are hard to see.

6. The inscriptions in figures 1 and 2 must be made in the same way as in the text.

7. In figure 2 b, the graph of the curve may be missing, with blue color

Author Response

Response to Reviewer 2 Comments

We would like to thank the reviewers for giving us constructive suggestions which would help us both in English and in depth to improve the quality of the paper. Here we submit a new version of our manuscript with the title “Two Efficient Sparse Fourier Algorithms Using Matrix Pencil Method”, which has been modified according to the reviewers’ suggestions. Efforts were also made to correct the mistakes and improve the English of the manuscript.

Point 1: For the described algorithms, we ask the authors to give their decoding of the abbreviation of the name (for example, AAFFT, FFAST, DSFFT and others)

Response 1: Thank the reviewer for the comments. Your suggestion is greatly appreciated. We agree and therefore the correction has been made in Chapter 1.

Point 2: The authors use various algorithms such as sFFT1.0, sFFT2.0, sFFT3.0, sFFT4.0, sFFT5.0, and there is confusion in the presented material in the article. Is it possible for the authors to indicate in lines 231 and 242 references to the relevant literature.

Response 2: Thank the reviewer for the comments. Your suggestion is greatly appreciated. We agree and therefore have annotated relevant literature in the corresponding paragraphs.

Point 3: Is it possible for the authors to create a table of comparative representation of the formulas of all presented algorithms?

Response 3: Thank the reviewer for the comments. Your suggestion is greatly appreciated. In Chapter 5, a summary table to compare the performance of the proposed algorithms and the related algorithms has been added.

Point 4: Why in chapter 5.1. Experimental Setup missing comparison with sFFT3.0 algorithm ?

Response 4: Thank the reviewer for the comments. The sFFT3.0 algorithm, FFAST algorithm and sFFT-DT algorithm are ignored because they are not suitable for general sparse case, only suitable for exactly sparse case. This reminder has also been added in the article.

Point 5: Drawings 1 and 2 should be rebuilt due to the fact that they are hard to see.

Response 5: Thank the reviewer for the comments. Your suggestion is greatly appreciated. All figures have been rebuilt for the better display quality.

Point 6: The inscriptions in figures 1 and 2 must be made in the same way as in the text.

Response 6: Thank the reviewer for the comments. Your suggestion is greatly appreciated. We agree and therefore the inscriptions in figures have been modified to be consistent with the content in the text.

Point 7: In figure 2 b, the graph of the curve may be missing, with blue color

Response 7: Thank the reviewer for the comments. The DSFFT algorithm under the condition of large sparsity is ignored because it cannot deal with large sparse data. This reminder has also been added in the article.

Round 2

Reviewer 1 Report

The authors have addressed all of my comments and the paper can now be accepted after considering following modification.

-        Kindly modify and correct the font and size of the texts in the added block diagram in Figure1.

Author Response

Response to Reviewer 1 Comments

We would like to thank the reviewers for giving us constructive suggestions which would help us both in English and in depth to improve the quality of the paper. Here we submit a new version of our manuscript with the title “Two Efficient Sparse Fourier Algorithms Using Matrix Pencil Method”, which has been modified according to the reviewers’ suggestions. Efforts were also made to correct the mistakes and improve the English of the manuscript.

General Comments

Point 1:  Kindly modify and correct the font and size of the texts in the added block diagram in Figure1.

Response 1: Thank the reviewer for the comments. We have modified and corrected the font and size of the texts in the added block diagram in Figure1.